# Metric Embedding Learning on Multi-Directional Projections

**Gábor Kertész** 

John von Neumann Faculty of Informatics, Obuda University, 1034 Budapest, Bécsi út 96b, Hungary; kertesz.gabor@nik.uni-obuda.hu

**Abstract:** Image based instance recognition is a difficult problem, in some cases even for the human eye. While latest developments in computer vision—mostly driven by deep learning—have shown that high performance models for classification or categorization can be engineered, the problem of discriminating similar objects with a low number of samples remain challenging. Advances from multi-class classification are applied for object matching problems, as the feature extraction techniques are the same; nature-inspired multi-layered convolutional nets learn the representations, and the output of such a model maps them to a multidimensional encoding space. A metric based loss brings same instance embeddings close to each other. While these solutions achieve high classification performance, low efficiency is caused by memory cost of high parameter number, which is in a relationship with input image size. Upon shrinking the input, the model requires less trainable parameters, while performance decreases. This drawback is tackled by using compressed feature extraction, e.g., projections. In this paper, a multi-directional image projection transformation with fixed vector lengths (MDIPFL) is applied for one-shot recognition tasks, trained on Siamese and Triplet architectures. Results show, that MDIPFL based approach achieves decent performance, despite of the significantly lower number of parameters.

**Keywords:** deep metric learning; one-shot learning; multi-directional image projections; object matching; object re-identification

## 1. Introduction

The popularity of deep learning was fueled by image recognition. In the early 2010's, participants on the ImageNet Large Scale Visual Recognition Challenge (ILSVRC) [1] started to use Convolutional Neural Networks (CNN) with a large number of hidden layers to process images. In just a few years, performance and efficiency exploded due to multiple factors [2], including the ability to use GPUs for training, and the introduction of novel algorithms to solve the vanishing gradient problem of hidden layers. In every year of the challenge, error rates decreased significantly, while new architectures, revolutionary ideas were introduced.

The goal of the ILSVRC was to create a model that could accurately classify images into a large number of classes—the exact number of categories was 1000. The highest performing solutions were all based on the same structure: a deep neural network with convolutional filters, which are wired onto fully connected layers. The output of the network is a layer with an element number equal to the number of classes. These solutions work well in case the number of classes are fixed. In the case that the model needs to be extended to be able to identify instances from newly added categories, the network architecture would need restructuring and for the parameters to be retrained.

In case the problem is instance recognition instead of categorization, where instance numbers are unknown, the method of matching is different. The Siamese Neural Network (SNN) is a two-headed structure, where the inputs are flown through the same network, and mapped to vectors of a

multi-dimensional space. The semantic distance of the vectors is measured [3], and similarity is given. The main idea is that the images of the same instances are mapped to vectors whose distance are minimal, while the representations of different objects are distant. In recent years, modern Siamese applications [4], and novel triple-headed structures have also been introduced [5]. As the clusters of elements define categories, instance classification could be done by finding the nearest neighbours. Distance calculation and cluster retrieval significantly increases the computational complexity of the problem.

Feature extraction based on image data mostly relies on the RGB (Red-Green-Blue) representation of pixel information. While deep CNNs on raw image data provide high performance, the parameter number of the model results in high memory cost. This memory cost is easily handled by GPU-enabled workstations, however, implementation on wearable, small hardware is impossible. As an efficient solution, the MobileNet [6] architecture was proposed, which produced high performance on the ImageNet dataset and had only a small number of parameters. In the last years, applications based on this architecture became very popular [7], which is primarily motivated by the portability of the model.

In these cases, a server-client architecture is proposed, where the clients transfer the input parameters to the server, that then makes the prediction, and replies with the class identifier [8]. In the case of IoT devices, transfer load of large images could cause bottleneck in processing. To tackle this challenge, researchers proposed a structure where data transfer is reduced between nodes and the central workstation [9].

Principal Component Analysis (PCA) is a technique which extracts the dominant patterns of a data matrix, resulting in the compression of the input data [10]. PCA based compression of images is feasible, although the processing time and feature selection is questionable [11]. Other methods for dimensionality reduction also exist [12], which could be categorized as linear or non-linear, describing the relationship between the input data and the extracted features. A neural network based approach for PCA could be the use of the autoencoder [13] to reduce the dimensionality of data. The application in machine learning tasks results in efficient solutions for classification and regression [14]; recent applications are based on the ability to remove the noise and redundancy of the processed data [15]. Other methods based on low-level imaging can also produce a compression of the data, with a reasonable computational time [16–18].

The method proposed in the following sections is based on Multi-Directional Image Projections with Fixed Length (MDIPFL) [19]. Input images are transformed into projection space, and these matrices are transferred to the deep neural network. The model is trained on projection data with triplet samples to have optimal discriminative abilities. As image sizes decrease, the model parameter number can be reduced as well, resulting in the centralized topography being eliminated.

The remainder of this paper is structured as follows: Section 2 presents a literature review of relevant techniques on metric learning and imae projection transformations. Section 3 introduces the proposed novel method on MDIPFL based embedding learning. Section 4 presents the results of two experiments, to show the discriminative ability of the method, as well as the one-shot classification performance in a real application. Section 5 discusses the results, and finally Section 6 summarizes the findings and presents possible future research topics.

## 2. Related Work

Since the introduction of Convolutional Neural Nets [20], the technique has undergone revolutionary transformations. Novel architectures on a large number of hidden, deep layers have been introduced to improve representational learning [21–23], while fundamental design patterns [24] and best practices [25] have been presented.

In metric learning, the Siamese structure was the first CNN-based approach to measure the similarity of images [3]. In the following years, novel ideas and applications have emerged [4,26,27]. Applications on facial recognition [28] gained popularity, as error rate dropped below a level where production-level implementation became possible.

The basis of the Siamese structure (visualized on Figure 1) is that a model with two consecutive sub-networks is applied. The two identical models are responsible for feature extraction on each of the inputs, and the output is given as the measured distance between the two encoded representations.

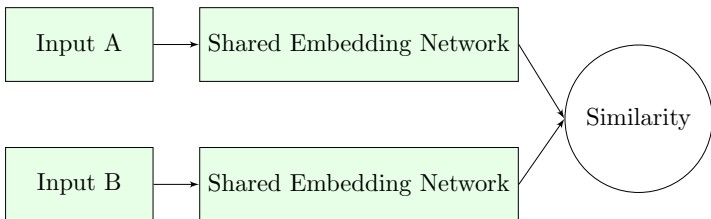

**Figure 1.** The architecture of the Siamese Neural Network. The underlying backbone network is shared, the output of the structure is a similarity score.

The output is binary and represents the similarity of the two. The loss is defined as contrastive loss [29], where the distance of the encodings of true pairs are reduced, and at the same time, negative pairs are separated and distance is increased. The contrastive loss function is defined as:

$$\mathcal{L}_{\texttt{contrastive}} = (1 - Y)\frac{1}{2}\texttt{max}(0, (m - d(x_1, x_2))^2) + Y\frac{1}{2}d(x_1, x_2)^2, \tag{1}$$

where $x_1$ and $x_2$ are the two inputs, $Y$ represents the target label, $m$ stands for the margin, and $d(a, b)$ defines the distance between given parameters.

As the expected labels are in codomain $\{0, 1\}$ and the two parts of the equation are distinguished by the $Y$ and $1 - Y$ multipliers. If $x_1$ and $x_2$ are different objects, $Y = 0$, then the loss is defined as the non-negative squared distance between the two and is eased by a margin value. If the two elements are similar, $Y = 1$, then the loss is given by the squared distance. This constant reduction of positive distances results in small clusters. This equation also gives, that for negative pairs, the loss is only positive when the distance is lower than the value defined as the margin. Therefore, the margin stops the enlargement of cluster distances.

The pairwise loss is referred to as contrastive as optimization is done based on positive or negative pairs. To increase performance, a novel loss function was introduced with FaceNet [5]. Triplet loss is based on three samples: an anchor, a positive, and a negative pair. The objective is to push negative pairs compared to positive pairs. Formally, triplet loss can be given as:

$$\mathcal{L}_{\texttt{triplet}} = \texttt{max}(0, m + d(x_a, x_p) - d(x_a, x_n)), \tag{2}$$

where $x_a$ represents the anchor and $x_p$ and $x_n$ stand for the positive and negative pairs, respectively.

As the loss is given by the difference between the negative distance and the positive distance with margin included, every case is penalized where $m + d(x_a, x_p) > d(x_a, x_n)$. In other cases, loss is 0. As a general result of this definition, positive pairs are kept together, while negative pairs are pushed away, within the frames defined in margin $m$.

The architecture of the triplet loss based neural network is different from the two-headed Siamese architecture, as the output is the encoding of the given input. Therefore, it is an encoder-type structure, where the training is done by measuring the distances of the encoding vectors, and classification is done by finding the closest neighbours in the embedded space. A diagram on the structure is shown on Figure 2.

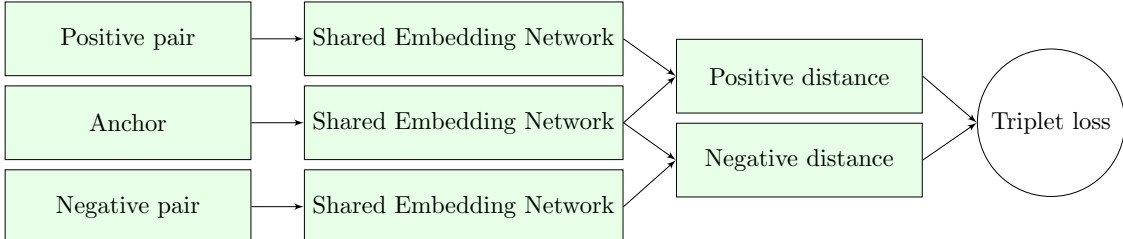

**Figure 2.** The architecture of a triplet based embedding network. The embedding model is the same between the three samples, the output encoding is used for metric-based classification.

During training, the samples given to the network should be chosen wisely. In ideal cases, for every sample, the farthest positive and the closest negative pair should be considered to have significant loss in order to find the best representation. Triplet mining [30,31] is an important step to select triplets for training batches [32]: those triplets where loss is zero would not help and therefore, should be ignored. However, triplets with the highest loss are computationally expensive to find, especially the selection of the closest negative pair is intense, as all encodings from different classes should be compared against the anchor element.

It is worth mentioning, that similar solutions were introduced by Zhu et al. [33] in person re-identification. The image-video triplet method–which is based on the same fundamental idea of comparing three samples–was used in an approach on multi-view person re-identification to improve the discriminative ability of the model.

In the paper describing OpenFace [34], a novel approach to mine batches of triplets for efficient training is introduced. While so-called *offline* triplet mining would have serious overhead, an *online* method of selecting triplet with non-zero loss on-training provides a significant speedup. A difference in selecting multiple negatives (known as the batch-all strategy) and calculating the mean error results in less of an effect compared to the so-called batch-hard strategy that takes the *hardest* positive and negative pair from the batch and results in the highest loss [35]. Recent results show, that easy-positive triplet selection causes flexible embedding space representation and has significant results in performance [36].

To map an image to its projections, multiple possible transformations can be applied. While the earliest published analysis of 2D mapping to 1D projections was done by Radon in the early 20th century [37,38], the applications on a similar solution by Hough [39] became de facto standards in the image processing era for projection-based operations [40]. The Radon transform was rediscovered [41], and became popular especially because of the inversion formula [42]. Beyond the applications in the field of medicine, applications in image object matching [43–45] are also popular.

A similar mapping to multi-directional projections is the MDIPFL transformation [19]. MDIPFL uses a fixed number of bins regardless of the projection length and rotational angle. The result of this transformation is a matrix containing different projection profiles, with a fixed size independent from the input image size. Therefore, the bin number variable directly affects the compression of the method. Samples of the Radon and MDIPFL transformations are visualized on Figure 3.

As the output of the MDIPFL transformation is independent from the input image size, the commonly applied resizing of the input image is unnecessary. The transformation is highly parallelizable [17] and implementation on IoT devices is possible. Considering that the memory cost is also reduced, the effects on transfer overhead are reduced, therefore, the implementation in environments with heavy communication, e.g., distributed smart camera networks [46–48] is feasible.

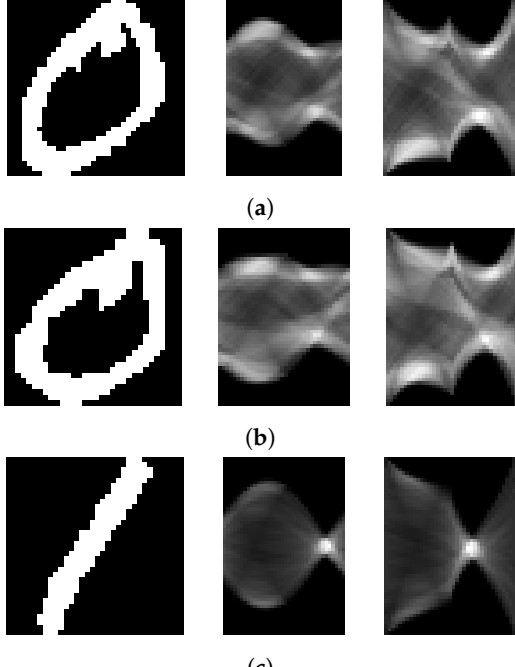

**Figure 3.** Samples of the Radon and MDIPFL transformations: from left to right the original images, the Radon transformations and the MDIPFL transformations, respectively. Both projection transformations show the projection profiles on range $[0; \pi]$. The original images are samples from the NIST [49] dataset. Subfigure (**a**,**b**) show observations from the same class, subfigure (**c**) is different.

## 3. Methodology

To ensure that the object signatures can be transferred among smart camera network nodes, a transformation with low computational and memory cost should be applied on the device. These signatures should be compared in the same place afterwards, therefore, the trained neural comparator should be cost-efficient.

In the following, the transformation that maps an image to its multi-directional projections, reducing the input size, is further introduced. The training method of the model which encodes the input to representing embedding vectors is defined in detail. In this paper both contrastive and triplet loss based architectures are analyzed. Re-identification of instances is done by measuring the distance in embedding space and model evaluation is done by calculating the one-shot classification accuracy.

### 3.1. Multi-Directional Projections

As defined in the paper introducing the MDIPFL transformation [19], the length of the output projection functions are equal and adjustable by setting the so-called bin number. Therefore, the sinusoid effect, observable in Radon space and which is caused by projecting rectangular shapes, is eliminated.

The size of the output matrix is $N \times \lceil \frac{\pi}{\texttt{Step}} \rceil$, where $N$ stands for the bin number, and $\texttt{Step}$ represents the rotation increment after each projection. Rotations are interpreted in range $[0; \pi]$, as rotations in $[\pi; 2\pi]$ are exactly the same, but mirrored [50]. The intensity values of the elements are summed, including the partially covered elements. The method is visualized on Figure 4.

The computational complexity is only dependent on the input image size and the $\texttt{Step}$ variable. parallel processing is possible, as the calculation of each projection angle is independent. Geometric decomposition based parallel execution is also possible, making GPU-enabled data-parallel implementation possible [51]. For further details on the algorithm and possible implementations, refer to our earlier work [19].

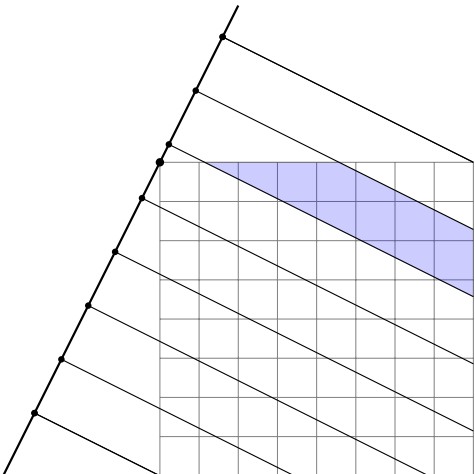

**Figure 4.** Visualization of the MDIPFL transformation. Intensity values of covered elements are summed, in the rate of coverage.

### 3.2. Pre-Training

In similar metric-based learning problems, researchers often apply a multi-class pretraining of the model to set the initial parameter values efficiently, and to achieve high overall performance [52–54]. As an initial step, the selected base model is extended with a final, fully-connected layer with an element number equal to the number of training categories and uses softmax activation function and cross-entropy as loss. The Adam [55] optimization algorithm is known for high performance in training of multi-class classifiers, application is based on empirical results. Other design decisions, such as the linear learning rate decay and rate reduction based on loss convergence, are also well-known techniques to improve performance [25].

Dealing with overfitting is important in the case of pretraining. By monitoring the calculated loss, training can be shut down when convergence stops. The model with the highest classification accuracy is restored, and processing continues with metric training.

### 3.3. Embedding Learning

As previously stated, metric based similarity learning can be done using a Siamese architecture, or a Triplet Network. The contrastive loss of the SNN reduces the distance between the representations to become minimal, while the triplet loss based on three samples creates clusters of similar observations.

To optimize performance, the mining of negative samples is key [35] and three strategies can be distinguished:

- random mining;
- semi-hard mining;
- hard mining.

The ideal course of action would be to train the model with triplets where each anchor is accompanied by the ideal positive and negative pairs. It is evident, that the selection of the farthest positive pair is feasible, however, selecting the closest negative pair is computationally intense, as all possible negatives have to be checked.

According to this, random mining is the most efficient, as randomly selecting the negative pair is simple; these negatives are often referred to as *easy* negatives. However, as the distance between the negative pair and the anchor could be higher than the distance between the positive pair and the anchor, calculated loss could yield zero (as given in (2)).

To solve this issue, semi-hard negatives should be mined. A negative pair is considered *semi-hard* when the difference between the negative distance and the positive distance is less than the margin.

In this case, the positive pair is actually closer to the anchor, but the marginal distance results is a positive loss. While the mining is more intense, the positive loss results in convergence.

The most intense method to mine negatives, referred to as *hard* mining, is done by selecting those negatives that are closer to the anchor than the positives. These different categories of triplets are visualized on Figure 5.

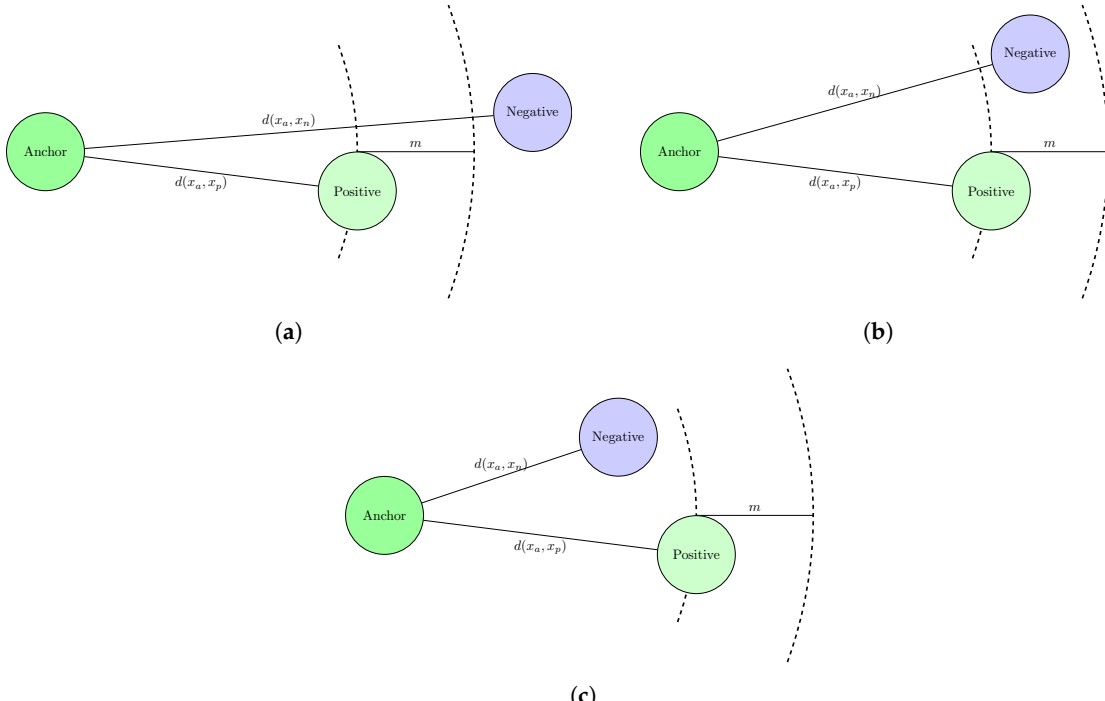

**(a)**  **(b)**

**(c)**

**Figure 5.** Illustration of different types of triplets. Subfigure (**a**) shows an easy negative, where $d(x_a, x_n) > d(x_a, x_p) + m$, resulting in zero loss. Subfigure (**b**) illustrates a semi-hard negative, as $d(x_a, x_p) < d(x_a, x_n) < d(x_a, x_p) + m$, resulting in a positive loss. On subfigure (**c**), the so-called hard negative is visualized, where $d(x_a, x_n) < d(x_a, x_p)$.

The triplet mining method introduced in OpenFace [34] is efficient when compared to so-called offline mining, as a batch of samples are pre-selected, all triplets are generated, and invalid triplets are masked.

### 3.4. Measuring Performance

On defining the performance of the novel projection based embedding learning method, one-shot classification accuracy is applied. In comparison to k-NN (k-nearest neighbors) [56], one-shot classification is done by selecting a subset of all available samples, therefore, the scale is relative, and accuracy represents the separational capabilities.

The method measuring one-shot classification accuracy is described in several papers [4,57]. A brief informal summary of the algorithm for $N$-way classification is as follows:

1. an anchor observation is randomly selected;
2. a true pair is selected from the same category;
3. $N - 1$ false pair samples are selected from other categories;
4. distances between the anchor and pairs are measured;
5. classification is marked correct, if the distance between the anchor and the true pair is minimal amongst other anchor-based distances;
6. steps 1–5 are repeated $k$-times, while correct classifications are counted;
7. $N$-way one-shot classification accuracy is given as $\frac{\text{Correct classifications}}{k}$.

In the experiments explained in the next section, both Siamese and Triplet structures are evaluated on different sets of data, with and without pretraining, and by applying semi-hard and hard negative mining for a given number of selected classes.

## 4. Results

To compare the discriminative ability of projection-based and raw-image based metric learning, comparative analysis of these methods should be done. To do so, a public dataset with relatively simple samples with a small parameter number is necessary, while the number of categories and instances are high enough to apply embedding distance based classification.

### 4.1. Discriminative Ability

The popular MNIST dataset [58] contains images of handwritten digits, with image sizes fixed in $28 \times 28$. For 60,000 pre-selected training images, 10,000 test images are also provided. The EMNIST (extended MNIST) dataset [59] also includes letter representations among the digits, therefore, the number of categories is increased, while image size is reduced to $28 \times 28$. The data of both MNIST and EMNIST originates from the NIST dataset [49]. This database contains the data from 3669 handwritten forms, where digits and letters are segmented, labeled, and transformed to a $128 \times 128$ size. The total number of characters is 814,255.

For the first experiments, the NIST dataset was applied. While the set contains imbalance between the classes regarding the number of samples, this will be handled during training by selecting a subset of samples with an equal size from each classes. As recommended by the dataset descriptor [49], extracted records from handwriting sample form nr. 4 is selected as the validation set.

As a preprocessing step, the original images are cropped, the region of interest is highlighted, and they are extended to be square. Before MDIPFL transformed images are inverted, the result is images where the background pixel intensities are zeros, and foreground content information is represented as positive integers. Samples are shown on Figure 6.

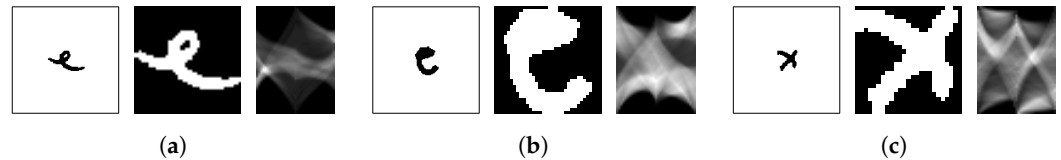

(a)                (b)                (c)

**Figure 6.** Samples of the NIST dataset before and after content highlighting, and MDIPFL transformation. Subfigures (**a**,**b**) represent character lowercase *e*. Subfigure (**c**) shows a sample of lowercase *x*.

As an overall experiment, multiple setups are examined, including the original NIST datarecords with size $128 \times 128$, the cropped images resized to $96 \times 96$, and the results of the MDIPFL transform that are sized $50 \times 37$. The neural architectures applied are custom CNNs, where the design decisions are inspired by the VGG architectures [24]. The selected encoding vector length is 64. The structures are constructed so the number of trainable parameters are similar, around $6M$, in the case of the original NIST set, $9M$.

For training, 1500 images from a total of 62 classes are selected. Where multi-class classification pretraining is applied, a maximum of 10 epochs are done with 100 steps per epoch, and 32 samples in each step. During training, a maximum number of 1000 iterations with 500 steps is set. If, however, convergence fails for 10 consecutive steps, training is shut down. As the optimizer, the novel rectified Adam algorithm [60] is applied.

The architectures were implemented in Keras [61] over Tensorflow [62], and by using the EmbeddingNet framework [54]. In the spirit of open access and reproducible results, the source

codes (including a batch of useful scripts), configurations, and result logs are made publicly available at https://github.com/kerteszg/EmbeddingNet as supplementary materials.

Table 1 presents the measured one-shot classification accuracy of each method on the validation subset. Based on the results, multiple conclusions can be done. Firstly, embedding learning in MDIPFL space is feasible as the learned representations are suitable for discrimination. Furthermore, the application of few epochs of multi-class pretraining have a stabilizing effect on performance.

**Table 1.** The performance of each method for metric learning on the NIST dataset, given by the one-shot classification accuracy on the validation subsets. 1000 classification tasks were evaluated for each measurement.

| | Pretraining | Architeture | Mining | N-Way | | | | | | | | | |
|---|---|---|---|---|---|---|---|---|---|---|---|---|---|
| | | | | 1 | 2 | 3 | 4 | 5 | 6 | 7 | 8 | 9 | 10 |
| Original 128 × 128 | Yes | Siamese | | 100.0 | 98.3 | 96.4 | 95.6 | 93.7 | 92.2 | 90.9 | 90.4 | 88.8 | 89.9 |
| | Yes | Triplet | Semi-hard | 100.0 | 94.8 | 87.7 | 86.0 | 83.9 | 80.5 | 82.0 | 75.6 | 77.5 | 74.2 |
| | Yes | Triplet | Hardest | 100.0 | 95.7 | 92.5 | 91.1 | 86.7 | 82.4 | 81.0 | 80.2 | 76.9 | 75.3 |
| | No | Siamese | | 100.0 | 98.2 | 97.5 | 94.9 | 94.7 | 92.3 | 93.6 | 90.1 | 87.8 | 89.5 |
| | No | Triplet | Semi-hard | 100.0 | 94.7 | 93.4 | 88.0 | 86.7 | 83.5 | 83.3 | 79.8 | 80.1 | 76.7 |
| | No | Triplet | Hardest | 100.0 | 95.8 | 91.0 | 87.1 | 87.7 | 84.5 | 80.4 | 80.6 | 78.3 | 76.1 |
| Highlighted 96 × 96 | Yes | Siamese | | 100.0 | 97.5 | 94.7 | 94.0 | 94.5 | 90.5 | 91.6 | 89.7 | 88.0 | 86.5 |
| | Yes | Triplet | Semi-hard | 100.0 | 90.3 | 83.8 | 81.3 | 77.3 | 72.4 | 68.2 | 68.2 | 65.1 | 60.4 |
| | Yes | Triplet | Hardest | 100.0 | 94.1 | 86.3 | 85.9 | 81.2 | 78.1 | 77.6 | 71.7 | 68.1 | 69.8 |
| | No | Siamese | | 100.0 | 94.9 | 92.8 | 91.1 | 88.8 | 84.9 | 83.3 | 84.4 | 82.1 | 79.8 |
| | No | Triplet | Semi-hard | 100.0 | 81.9 | 70.3 | 65.7 | 61.4 | 57.2 | 52.1 | 49.1 | 45.0 | 46.3 |
| | No | Triplet | Hardest | 100.0 | 90.4 | 83.0 | 80.5 | 76.4 | 68.9 | 70.6 | 66.1 | 64.2 | 64.2 |
| MDIPFL50 50 × 37 | Yes | Siamese | | 100.0 | 97.3 | 95.2 | 94.0 | 90.4 | 88.8 | 87.1 | 88.3 | 87.2 | **86.2** |
| | Yes | Triplet | Semi-hard | 100.0 | 92.4 | 87.3 | 86.2 | 81.5 | 77.6 | 75.0 | 76.0 | 69.6 | 70.6 |
| | Yes | Triplet | Hardest | 100.0 | 96.2 | 93.2 | 88.3 | 87.1 | 82.9 | 79.9 | 81.1 | 79.5 | 75.1 |
| | No | Siamese | | 100.0 | 97.7 | 95.1 | 93.9 | 92.7 | 92.2 | 88.0 | 88.1 | 85.9 | **87.3** |
| | No | Triplet | Semi-hard | 100.0 | 80.8 | 72.4 | 66.4 | 61.0 | 54.7 | 54.8 | 50.6 | 45.6 | 42.2 |
| | No | Triplet | Hardest | 100.0 | 94.3 | 91.8 | 87.3 | 83.4 | 80.8 | 79.3 | 77.3 | 77.2 | 75.5 |

Results also show, that Siamese architectures outperform the presented triplet-based structures. The possible cause of this phenomenon is the relatively small number of classes, where calculated contrastive loss is more effective than the triplet loss technique. As the measured triplet loss fails to improve over time, early stopping shuts down training and results in a suboptimal learning of embedding vectors.

On Figure 7, the evolution of the measured loss during training is visualized. The effects of multi-class pretraining are shown on the subfigure (a) by visualizing the pretraining loss and the measured loss during metric learning. As clearly visible, all three applied models and inputs, including the projection profile matrices, are showing signs of similar behavior on model fitting.

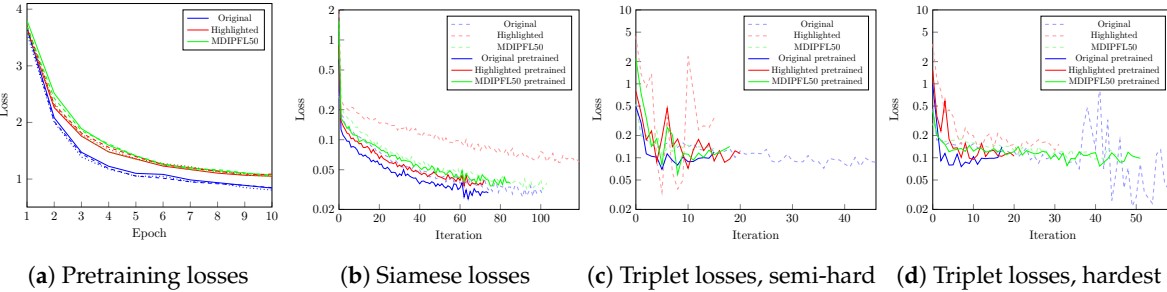

(**a**) Pretraining losses　(**b**) Siamese losses　(**c**) Triplet losses, semi-hard　(**d**) Triplet losses, hardest

**Figure 7.** The evolution of loss during different phases of training. Subfigure (**a**) illustrates the measured pretraining loss for the three different inputs and models. (**b**) visualizes the measured loss on Siamese structured training, while (**c**,**d**) shows the change of triplet loss over training iterations for semi-hard and hardest negative selection, respectively. It is notable, that (**b**–**d**) use a logarithmic vertical scale for a more detailed representation.

Plot (b) of Figure 7 shows the measured loss of Siamese architectured training with and without pretraining for all three inputs. Subfigure (c) and (d) illustrates the same for the Triplet networks, semi-hard and hard negative selections, respectively.

In almost all cases, the training of pretrained models is completed in less iterations. Further, convergence of the measured error is smooth, free of outliers. It can be concluded, that pretraining helps to reduce the possibility of volatility during training. The theory is that by learning the basic feature representations of observations and setting the initial parameter values accordingly, the possibility of suboptimal selections are reduced in the embedding learning phase.

When comparing the triplet nets with SNNs, it is clear that the iteration number until training is stopped is around ∼25% in favor of the triplet architecture, however, the measured contrastive loss is below the triplet loss. In the occurence of the loss of Triplet Networks, selecting the hardest negative pair slightly affects the calculated loss through the iterations.

While final performance is the key metric when comparing models, the time of training could also be a key factor. In Table 2, the training times for different experiments can be compared.

**Table 2.** The measured runtimes of the experiments for training on the NIST SD19. Each iteration took 500 steps. Times of pretraining not included.

| | Pretraining | Architeture | Mining | Training Time (Seconds) | Iterations | Time per Iteration (Seconds) |
|---|---|---|---|---|---|---|
| | Yes | Siamese | | 7488.01 | 75 | 99.84 |
| | Yes | Triplet | Semi-hard | 2528.65 | 16 | 158.04 |
| Original | Yes | Triplet | Hardest | 2907.28 | 18 | 161.52 |
| 128 × 128 | No | Siamese | | 10,263.05 | 102 | 100.62 |
| | No | Triplet | Semi-hard | 9047.23 | 47 | 192.49 |
| | No | Triplet | Hardest | 9720.34 | 60 | 162.01 |
| | Yes | Siamese | | 2976.65 | 73 | 40.78 |
| | Yes | Triplet | Semi-hard | 1399.09 | 21 | 66.62 |
| Highlighted | Yes | Triplet | Hardest | 1454.04 | 21 | 69.24 |
| 96 × 96 | No | Siamese | | 1984.10 | 120 | 16.53 |
| | No | Triplet | Semi-hard | 518.04 | 16 | 32.38 |
| | No | Triplet | Hardest | 1212.54 | 32 | 37.89 |
| | Yes | Siamese | | 2430.68 | 86 | 28.26 |
| | Yes | Triplet | Semi-hard | 890.85 | 16 | 55.68 |
| MDIPFL50 | Yes | Triplet | Hardest | 2500.35 | 52 | 48.08 |
| 50 × 37 | No | Siamese | | 3017.49 | 104 | 29.01 |
| | No | Triplet | Semi-hard | 888.62 | 19 | 46.77 |
| | No | Triplet | Hardest | 2425.68 | 46 | 52.73 |

Where pretraining was done, the total training time should be adjusted. In the completed simulations, multi-class pretraining took ∼50, ∼80, and ∼210 s for the 50 × 37, 96 × 96, and 128 × 128 inputs, respectively.

The general observation that the processing of less data can be done in less time is trivial, therefore, the projection based inputs are processed in minutes. Additionally, the table also shows that the Siamese networks are training for more iterations, although these iterations take slightly less time. Furthermore, the times per iteration also indicate small differences between hardest and semi-hard negative mining strategies.

### 4.2. Object Re-Identification

Metric embedding learning of image representations is often used for re-identification of people [63,64] or vehicles [65]. To analyze the performance of the model in a real-life application, vehicle images are used as inputs.

The dataset published in the International Workshop on Automatic Traffic Surveillance of CVPR 2016 [66] is processed, and the collected samples are used for the experiment. The two different viewpoints are presented on Figure 8.

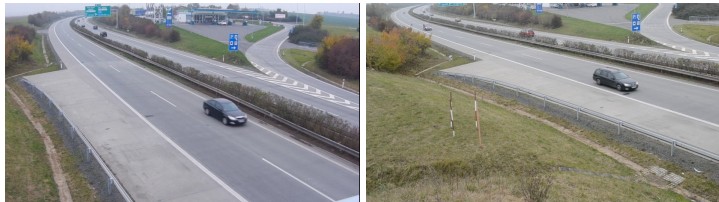

**Figure 8.** Sample images from the Automatic Traffic Surveillance Workshop on CVPR2016 [66]. All vehicles are recorded from two slightly different viewpoints. The dataset also contains annotation information that can be used to extract observations.

The records of the dataset are labeled and annotated, the extraction of the regions of interest was done accordingly. For every instance, a total of 30 observations from two different viewpoints are used. For those objects where the number of observations is inadequate, records are removed from the dataset. The remaining set is split into a training and a validation subset using a 9 : 1 ratio; the total number of training instances is 629, with 69 test classes. Samples of the prepared dataset are shown in Figure 9.

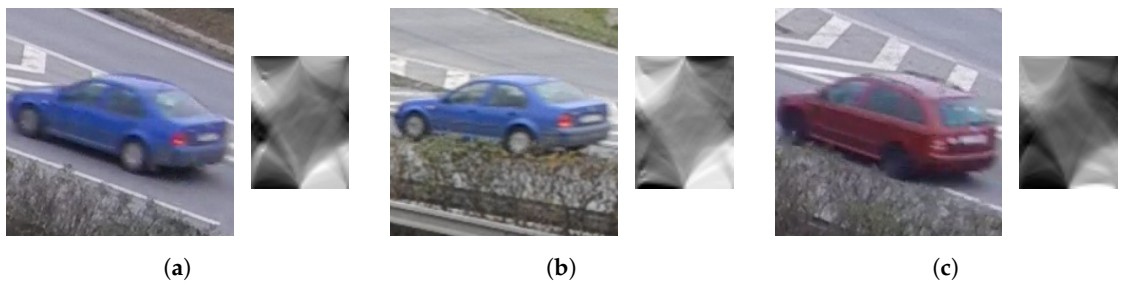

|      (a)      |      (b)      |      (c)      |

**Figure 9.** Samples of the dataset prepared using the data presented in [66]. Vehicle images are cropped to square, and transformed using the MDIPFL projection mapping. Subfigures (**a**,**b**) represent different observations of the same instance. Subfigure (**c**) shows a sample of a different instance.

As a backbone to the architecture, the smallest version of ResNet [23] is applied to map input images to encodings with a length of 128 elements. During training, a maximum number of 1000 epochs are done, with 200 steps per epoch. If pretraining is relevant, a maximum of 100 epochs with 500 steps on each is done. Early stopping could occur if loss convergence stops.

Measured one-shot classification accuracy is presented in Table 3. To correctly interpret these data, it is important to emphasize that the vehicle instances in the test set were not seen during training.

**Table 3.** The performance of each method for metric learning on the vehicle re-identification dataset, given by the one-shot classification accuracy. For each setup, 1000 one-shot classification tasks were evaluated on the test subset.

| Architeture | Pretraining | Mining | N-Way | | | | | | | | | |
|---|---|---|---|---|---|---|---|---|---|---|---|---|
| | | | 1 | 2 | 3 | 4 | 5 | 6 | 7 | 8 | 9 | 10 |
| Siamese | Yes | | 100.0 | 91.8 | 83.5 | 79.4 | 76.1 | 74.0 | 71.3 | 65.8 | 66.5 | 65.6 |
| Triplet | Yes | Semi-hard | 100.0 | 94.3 | 88.2 | 84.5 | 82.9 | 78.7 | 76.1 | 74.1 | 70.0 | 71.1 |
| Triplet | Yes | Hardest | 100.0 | 93.2 | 89.3 | 83.0 | 81.2 | 78.7 | 72.7 | 72.0 | 73.2 | 70.7 |
| Siamese | No | | 100.0 | 82.9 | 76.0 | 70.3 | 67.7 | 66.0 | 62.0 | 58.1 | 56.7 | 55.1 |
| Triplet | No | Semi-hard | 100.0 | 94.8 | 90.2 | 88.6 | 82.9 | 81.0 | 81.5 | 74.9 | 74.8 | **75.7** |
| Triplet | No | Hardest | 100.0 | 92.8 | 87.4 | 82.4 | 80.2 | 73.4 | 70.4 | 69.3 | 65.9 | 61.7 |

When training, the effects of pretraining are less significant. Figure 10 shows, that Siamese networks were able to reduce contrastive loss over time, while triplet loss, regardless of semi-hard or hardest negative mining, convergence of loss stopped around value ~0.1.

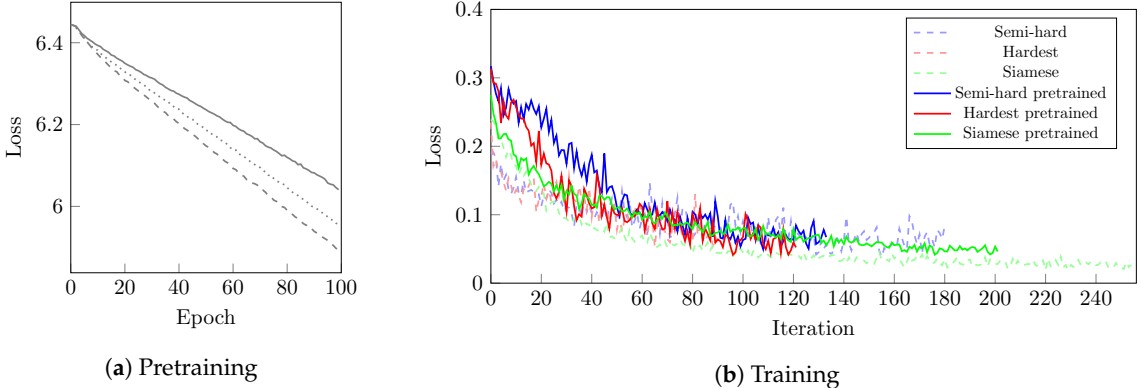

(**a**) Pretraining        (**b**) Training

**Figure 10.** Loss over time when training ResNet-18 using the MDIPFL50 transforms of the extracted samples of the ATS-CVPR2016 dataset. Subfigure (**a**) shows the linear pretraining loss, while subfigure (**b**) visualizes the contrastive and triplet loss over iterations for different setups.

It is notable, that while uninitialized SNN training took more time and steps when compared to the other methods, however all other approaches outperformed this method. The highest performance is measured when applying triplet architecture with semi-hard negative selection. While the hardest selection helps on the training data, semi-hard mining generalizes training effectively. On 10-way one-shot classification, the model trained only with triplet loss on semi-hard negatives achieves a decent performance of 75.7%.

As a further representation on model performance along with one-shot classification accuracy, t-SNE (t-Distributed Stochastic Neighbor Embedding) [67] is applied to reduce embedded vector dimensionality into 2D space. Figure 11 exhibits the effectiveness of the proposed method. As a baseline, t-SNE is also applied to an untrained network, resulting in randomly distributed points.

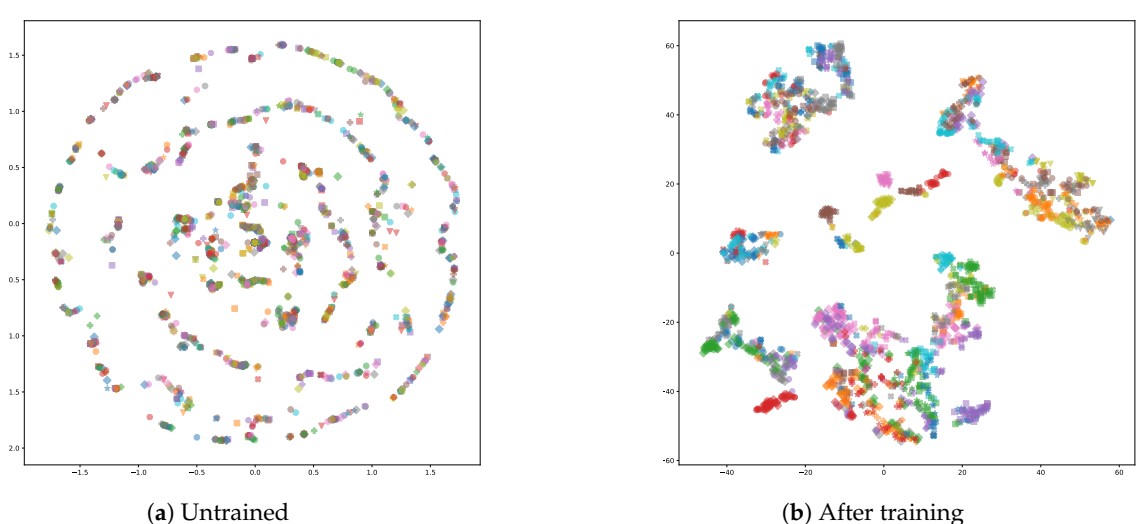

(**a**) Untrained        (**b**) After training

**Figure 11.** t-SNE [67] based visualization of the 128-dimensional embedding vectors from the test dataset in two-dimensions. Subfigure (**a**) illustrates the point-cloud before training, while subfigure (**b**) visualizes it afterwards. For the transformation perplexity was set to 30 and the learning rate to 200.

## 5. Discussion

Appearance based object re-identification is a classic problem, modern solutions of the deep learning era show great performance. Nowadays, with smart camera devices, on-edge solutions with a less memory needs are necessary. To reduce the parameter number, eighter the input or the model is shrinked, both having negative effects on performance. For a more efficient memory-performance tradeoff, input dimensionality reduction could be applied; these feature extraction methods has a significant computational complexity. In this paper, a multi-directional projection based method is presented to compress input data.

During representational learning in projection space, a shallow architecture is capable to distinguish different instances on a distance-based metric.

The results of the first experiments showed, that the MDIPFL transform based method has similar discriminative abilities as the original image based encoders when under similar circumstances, while the number of input parameters is reduced significantly.

The classification performance measured during the first experiment also resulted in the Siamese structure based training outperforming the triplet loss-based architecture. This was caused by the converging contrastive loss of SNNs, which led to more iterations, while the stagnating triplet loss led to early shutdown. Triplet loss is known to improve performance on a higher number of categories.

A second experiment was performed to confirm the applicability of the method for object re-identification. After evaluating the results of the application on the ATS-CVPR2016 dataset, it was concluded that while contrastive loss was reduced over iterations, triplet structure based embedding training outperformed the Siamese architecture approach.

It is worth to mention, that during training the contrastive loss of SNNs reduced loss over time, therefore, the validation loss based early shutdown of training to prevent overfitting happened later in contrast to Triplet nets. However, longer training did not result in higher performance, which is explainable by the behavior of constrastive loss for same class instances given earlier in Equation (1).

Another important conclusion is done regarding multi-class classification pretraining of the models: when the initial parameters were set during a short initial training, the final performance is more stable.

General observations following the experiments include the confirmation of multi-class pretraining usefulness. When the initial parameters were set during a short softmax-based training, the final performance is more stable. It was observed, that evolution of the calculated loss over iterations showed less volatile behavior. In conclusion, the initialization of parameters on learning basic feature representations with softmax, the possibility of diverging from optima is reduced.

To summarize the results, the presented method showed decent performance on 10-way one-shot classification on the validation subset of the dataset. It is trivial, that the processing of less data takes less time, however it is worth mentioning that the overhead cost of pretraining and transformation is not significant, and as parallel processing is applicable, cost reduction is conceivable. A more important aspect of interpreting the results is by expecting these projection-based object signatures to be transferred through network. As the parameter size is significantly reduced, the transfer cost is lower, therefore application in distributed environment is efficient.

## 6. Conclusions

In this paper the MDIPFL transformation for image dimensionality reduction was applied to compress input data of embedding learning methods. Multiple experiments were performed for instance recognition using this novel metric learning method including the Siamese architecture and triplet loss based methods. Multiple setups were examined, as the effects of multi-class pretraining and different negative selection methods were evaluated.

Results showed, that the proposed method is efficient in object re-identification tasks. The effect of multi-class classification pretraining for training stability were verified by multiple experiments. As the

compression reduces object signature length, it also supports application in low memory systems, implementation in distributed smart camera networks is possible.

The contribution of the paper could be extended with several future works, including the analysis of different representation-learning architectures, including NASNet [68–70], which is well-known for outperforming state-of-the-art methods for image classification tasks.

The presented projection mapping transformation could be compared to other dimensionality reduction methods, such as classic PCA or LDA (Linear Discriminant Analysis) [71–73]. These methods should be evaluated in terms of classification accuracy as well as compression ratio and transformation runtime complexity when applied as a compression method in metric learning.

As the performed experiments were executed to provide a proof-of-concept on projection based embedding learning, in the case of deployment in production, hyperparameter tuning [74] should be applied.

It is worth mentioning, that the application of transfer learning seems promising, as feature extraction with pre-trained layers could result in higher overall performance with less necessary training, and novel results show [75], that high learning performance could be achieved.

**Supplementary Materials:** The source codes and materials are made publicly available at https://github.com/kerteszg/EmbeddingNet.

**Funding:** This research received no external funding.

**Acknowledgments:** The research presented in this paper was carried out as part of the EFOP-3.6.2-16-2017-00016 project in the framework of the New Széchenyi Plan. The completion of this project is funded by the European Union and co-financed by the European Social Fund. The author thanks Sándor Szénási, Imre Felde, and members of the CUDA Teaching Center at Obuda University for constructive comments and suggestions.

**Conflicts of Interest:** The author declares no conflict of interest.

## Abbreviations

The following abbreviations are used in this manuscript:

| | |
|---|---|
| Adam | Adaptive Moment Estimation |
| CNN | Convolutional Neural Network |
| EMNIST | Extended MNIST |
| GPU | Graphics Processing Unit |
| ILSVRC | ImageNet Large Scale Visual Recognition Challenge |
| k-NN | k-nearest neighbors |
| MNIST | Modified NIST |
| MDIPFL | Multi-Directional Image Projections with Fixed Length |
| NIST | National Institute of Standards and Technology |
| PCA | Principal Component Analysis |
| RAdam | Rectified Adam |
| ResNet | Residual Network |
| SNN | Siamese Neural Network |
| t-SNE | T-distributed Stochastic Neighbor Embedding |
| VGG | Visual Geometry Group |

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
