# Peer review of "Metric Embedding Learning on Multi-Directional Projections"

_algorithms, doi:10.3390/a13060133_

Round 1
Reviewer 1 Report
Dear authors.
Thank you for the opportunity to contribute to your paper.
The following are some considerations for improving the paper.
Abstract.
The abstract has relevant information, but it remains to highlight the main results obtained in the paper.
Introduction.
The introduction of the article conceptualizes the theme, presents relevant contexts.
Highlight the paper's main contributions to science.
The (last) paragraph is missing, placing the organization of the paper presenting the reader with the other sections to be presented by the paper.
I advise changing items from 1.1 to 2.0. This factor is relevant to the bibliographic review of the paper. So check the rest of the numbering.
Avoid entering citations without the subject, as in In [32]. Please review the text.
2. Methodology
k-NN (k-nearest neighbors), but the quote for that job.
2 3. Results
Bold the best values ​​in each of the texts.
Separate the discussions from the conclusions.
Increase the discussions.
Please, in conclusion, put possible future works.
Highlight the positives and negatives of your approach.
Add more citations from 2019 and 2020 to identify state of the art on the topic.
Author Response
I would like to thank the reviewer for the careful reading of this manuscript and for the thoughtful comments and constructive suggestions, which helped to improve the quality. All suggestions were considered, the detailed responses are given below.
A revised version of the manuscript is submitted, along with a version where I have highlighted the changes.
Response to comments
Abstract.
The abstract has relevant information, but it remains to highlight the main results obtained in the paper.
Thank you for pointing this out. I agree that the abstract could be improved by adding details of the results, therefore, it has been slightly modified. However, I would like to add, that because of the 200 word limit of the abstract, this extension required some redesign.
Please refer to the abstract of the revised manuscript.
Introduction.
The introduction of the article conceptualizes the theme, presents relevant contexts.
Highlight the paper's main contributions to science.
The (last) paragraph is missing, placing the organization of the paper presenting the reader with the other sections to be presented by the paper.
Thank you for the valuable comments: I have extended the first section with a paragraph containing references to the further sections, shortly explaining the contents.
I advise changing items from 1.1 to 2.0. This factor is relevant to the bibliographic review of the paper. So check the rest of the numbering.
Thank you for this suggestion: the subsection has been moved, and became a separate section.
Avoid entering citations without the subject, as in In [32]. Please review the text.
I appreciate the recommendation: the selected citation, and all similar occurrences are fixed throughout the text.
2. Methodology
k-NN (k-nearest neighbors), but the quote for that job.
Thank you for pointing out the missing reference, it has been added.
2 3. Results
Bold the best values in each of the texts.
I agree that emphasizing the best values in the results helps evaluation. Table 1 and Table 3 are changed accordingly.
Separate the discussions from the conclusions.
Thank you for this suggestion: in the revised manuscript discussion and conclusion are separate sections.
Increase the discussions.
Based on the previous comment, discussion and conclusion has been separated, relevant parts restructured and extended.
Please, in conclusion, put possible future works.
Based on this valuable comment, the conclusion is further extended with future plans and possible further research directions.
Highlight the positives and negatives of your approach.
I agree with this feedback: the quality of the manuscript is greatly improved by a paragraph of overall evaluation of the proposed method. Therefore, this has been added to the conclusion section.
Add more citations from 2019 and 2020 to identify state of the art on the topic.
I appreciate this comment: based on the suggestion the last section is extended with references to current, state-of-the-art methods.
In addition to the above comments, all spelling and grammatical errors have been corrected. I look forward to hearing from the reviewer regarding this submission and to respond to any further questions and comments he may have.
Sincerely,
Gabor Kertesz

Reviewer 2 Report
I enjoyed reading this paper. Please find below a few minor suggestions:
- Please do not change the paragraphs too soon. Kindly merge the smaller paragraphs with the other.
- The following related papers are related to the image re-projection applied to different fields. Please discuss these papers in Section 1.1.
a) https://arxiv.org/abs/1606.02811
b) https://ieeexplore.ieee.org/document/8078250
- The code is available here: https://github.com/kerteszg/EmbeddingNet, I appreciate this. I'd suggest to include a few results (for example object re-identification) in the GitHub readme page.
- It will be interesting to see the discriminative ability using popular dimensionality reduction techniques: Principal component analysis (PCA) and Linear discriminant analysis (LDA).
Author Response
First of all, I am very grateful to the reviewer for the insightful and constructive comments on my paper. I am very thankful for the encouraging comments. All suggestions were considered, the detailed responses are given below.
A revised version of the manuscript is submitted, along with a version where I have highlighted the changes.
Response to comments
Please do not change the paragraphs too soon. Kindly merge the smaller paragraphs with the other.
I thank the reviewer for this comment, and I agree. Short paragraphs have been merged, as it is visible in the new version.
(Please refer to the new submitted version, as the document highlighting the differences shows the merged paragraphs separated, which is a well-known latexdiff issue (https://github.com/ftilmann/latexdiff/issues/118))
The following related papers are related to the image re-projection applied to different fields. Please discuss these papers in Section 1.1.
a) https://arxiv.org/abs/1606.02811
b) https://ieeexplore.ieee.org/document/8078250
Thank you for these suggestions: both papers are relevant to the discussed techniques, and are added to the first sections.
The code is available here: https://github.com/kerteszg/EmbeddingNet, I appreciate this. I'd suggest to include a few results (for example object re-identification) in the GitHub readme page.
First of all, I appreciate the positive feedback from the reviewer. All experiment results discussed in the manuscript are already available in the experiment-results directory (https://github.com/kerteszg/EmbeddingNet/tree/master/experiment-results). It was my original plan to show relevant excerpts of the manuscript on the page after publication, however based on the suggestion, I've added some details to the main page.
It will be interesting to see the discriminative ability using popular dimensionality reduction techniques: Principal component analysis (PCA) and Linear discriminant analysis (LDA).
Thank you for this suggestion. It would have been interesting to explore this aspect in terms of classification accuracy as well as compression ratio and transformation runtime complexity. However, as it is out of scope for this exact manuscript, I added it as a topic for further research.
In addition to the above comments, all spelling and grammatical errors have been corrected. I look forward to hearing from the reviewer regarding this submission and to respond to any further questions and comments he may have.
Sincerely,
Gabor Kertesz

Round 2
Reviewer 1 Report
Dear authors.
All of my suggestions were successful.
Reviewer 2 Report
All reviews have been successfully addressed.